# Understanding Professionals’ Knowledge Regarding Factors Influencing Changes in Attitudes toward Female Genital Mutilation/Cutting in Post-Migration Communities in Geneva, Switzerland

**DOI:** 10.3390/ijerph21060716

**Published:** 2024-05-31

**Authors:** Nasteha Salah, Nicola Cantoreggi, Patrick Petignat, Jasmine Abdulcadir

**Affiliations:** 1Institute of Global Health, University of Geneva, 1202 Geneva, Switzerland; nicola.cantoreggi@unige.ch; 2Division of Gynaecology, Department of Paediatrics, Gynaecology and Obstetrics, Geneva University Hospitals, 1211 Geneva, Switzerland; patrick.petignat@hcuge.ch

**Keywords:** female genital mutilation, female genital cutting, ecological model, cultural change, migration, abandonment of FGM/C, FGM, FGC, professional, knowledge, Switzerland

## Abstract

Female genital mutilation or cutting (FGM/C) is a practice involving the partial or complete removal of the external female genitalia for non-medical reasons. To facilitate attitude changes, the ecological model of behavior change considers multiple levels of influence and their relationships with environmental and behavioral factors. The combined effects of migration and cultural adaptation result in a transformative process that leads to decreased support for FGM/C. This qualitative study aimed to gain knowledge from FGM/C field professionals regarding the factors promoting behavioral changes in migrant communities in Geneva, Switzerland. Between September and October 2023, we conducted semi-structured interviews using a reflexive thematic analysis. Our qualitative research is reported in accordance with the COREQ criteria. A data analysis was performed using NVivo 14 software. Four influential dimensions were identified, each with associated factors. The first dimension, the social level, includes (1) the impact and implementation of anti-FGM/C laws. The second dimension, the community level, encompasses four factors such as (2) religion, (3) a multifaceted examination of social aspects, (4) navigating language barriers and raising awareness, and (5) cultural adaptation processes. The third dimension, the interpersonal level, includes factors such as (6) changing views on the marriage prerequisite. Finally, the fourth dimension, the personal level, is associated with (7) women’s experiences and perspectives regarding FGM/C. The findings highlight seven environmental factors, both within and across dimensions of the ecological model, that interact with human behavior to enable an adaptive cultural process. This process influences changes in attitudes and behaviors regarding FGM/C.

## 1. Introduction

### 1.1. Background

Female genital mutilation or cutting (FGM/C) is defined by the World Health Organization (WHO) as the partial or total removal of the external female genitalia for non-medical reasons [1]. This practice, which is cultural rather than religious, is outlawed in many countries where it is highly prevalent and in diaspora countries [2]. Nevertheless, it continues to prevail in Africa, Asia, and the Middle East; according to the 2024 UNICEF report, the numbers have been increasing, reaching 230 million girls and women worldwide [3,4]. In March 2024, lawmakers in Gambia voted to proceed with a measure aimed at repealing a ban on FGM/C that was implemented in 2015, potentially making Gambia the first nation to reverse protections against the practice [5]. As migration rates increase, the research suggests that communities in which FGM/C is traditional are also present in North America, Canada, Australia, New Zealand, and other Western nations [6,7]. There are over 600,000 women, living in at least 17 European countries, who have experienced FGM/C [8]. In 2023, it was estimated that 24,600 women and girls in Switzerland had experienced or were at risk of FGM/C [9]. Such indirect estimates are often used to measure the prevalence of FGM/C in diaspora countries without considering factors that could influence the behavior, attitudes, and beliefs of migrants regarding FGM/C.

### 1.2. Theoretical Framework Associated with Attitude Changes

In this study, we utilize the ecological model, which offers a holistic understanding of certain factors’ contribution to the formation of perspectives and norms related to the abandonment of FGM/C. It examines various dimensions—social, community, interpersonal, and personal [10]. The model provides insight into the processes of attitude change within communities originating from FGM/C-practicing countries in the context of migration. Cultural adaptation is characterized by the gradual adjustment of social, psychological, and cultural aspects, as ethnic groups align with societal norms over time. In the context of immigrant populations and the practice of FGM/C, measures of adaptation in a new setting include factors such as linguistic competence, the individual’s age upon arrival in the host country, the duration of residence, and the individual’s interactions within cross-border social networks [11]. Selective migration is the process in which individuals with higher levels of education, income, and urban residence are more likely to migrate. These individuals also appear to have lower support for FGM/C [12].

Numerous studies have investigated these interconnected factors associated with the ecological model, examining changes in attitudes and behaviors toward FGM/C. At the social level, the deterrent effect of anti-FGM/C legislation and the low number of reported FGM/C cases in many Western countries suggest that the practice may have been abandoned either before or after migration [13,14]. At the community level, in environments wherein the practice is unfamiliar and perceived negatively, individuals may feel less pressure to conform to the practice. Education campaigns and awareness-raising about the negative health effects may contribute to changing social norms and values that perpetuate the practice [13,15,16,17,18]. Additionally, the absence of a religious requirement and, at the interpersonal level, comparable marriage rates for uncut girls in communities compared to girls with FGM/C, can help question the necessity of the practice in this new setting [19]. Personal attitude changes involve breaking the taboos surrounding FGM/C and openly opposing the practice; these attitude changes are often prompted by introspection and the development of supportive social networks within the community [20]. The interaction between cultural adaptation, selective migration, and the ecological model can provide a deeper understanding of the mechanisms that shape changes in attitudes toward FGM/C cessation within post-migration communities [12,14].

### 1.3. Professionals in the FGM/C Field

Contemporary research on post-migration attitudes, beliefs, and knowledge regarding FGM/C in relation to healthcare providers has primarily centered on assessing the factual knowledge of healthcare professionals. This evaluation addresses several key aspects, such as the lack of knowledge, education, and training required to recognize and manage FGM/C-related needs. This deficiency can be attributed to inadequate guidelines, insufficient culturally sensitive approaches being provided, and various other social barriers that hinder the provision of FGM/C-related healthcare [21,22,23]. Moreover, most research on the impact of attitudinal change has been conducted at the community or individual level [13,15]. This qualitative study is the second article from a Ph.D. project which explores professional knowledge concerning factors contributing to attitudinal change toward FGM/C among post-migration communities in Geneva, Switzerland. The aim of this qualitative study was to explore how professionals in the field of FGM/C perceive and interpret the effects of migration and cultural adaptation regarding shifting attitudes about FGM/C in Geneva, Switzerland. 

## 2. Materials and Methods

### 2.1. Study Design and Data Collection 

Our qualitative study involved six women, with experience ranging from 7 to 28 years in the field of FGM/C. The participants were chosen through purposeful sampling [24]. This entailed choosing individuals with extensive knowledge and experience in the field of FGM/C based in Geneva, Switzerland. Two participants were from non-profit organizations: one a scholar, one an intercultural mediator, and two medical doctors specializing in gynecology and pediatrics. Participants were all proficient in either in French or English. We invited participants via email which included the study details, interview date, and location; reminders were sent two weeks prior to the interview and again two days before. 

Data collection occurred between September and October 2023 at the workplace, with three participants interviewed at the Geneva University Hospitals and two local non-profit organization settings. The final interview was conducted via Zoom software (version 5.17.11) with audio and video recording [25]. Four interviews were conducted individually and two simultaneously. Only the participant and the researcher (N.S.) were present during the interviews. The first author (N.S.) is a Ph.D. student at the University of Geneva, Switzerland, specializing in public health research focused on FGM/C. 

We conducted a semi-structured interview with participants and asked them to share their professional experiences, knowledge, and perspectives regarding shifts in attitudes toward FGM/C in the migrant communities residing in Geneva. The participants were encouraged to explore relevant topics related to the research question through open-ended questions [26]. We report our qualitative research in accordance with the COREQ criteria [27]. 

### 2.2. Interview Guide 

The interviews were conducted in French by the first author (N.S.) and had a duration ranging from 30 to 50 min. The structure of the interview questions was based on a framework derived from the reviewed literature on the factors associated with migration and attitude changes within communities originating from countries with a high prevalence of FGM/C. The semi-structured questionnaire was divided into three sections: The first delved into the origins and backgrounds of the population where FGM/C is prevalent; the second assessed the knowledge, and the third examined the attitude changes related to FGM/C in the Geneva context (Guide A1 in Appendix A). The questionnaire was not pilot-tested prior to data collection. We prompted critical thinking and encouraged the participants to consider the influence of migration and the post-migration context on the continuation of FGM/C or the factors that contribute to attitude changes in migrant communities. During each interview, we adjusted the question guide according to the participant’s input, encouraging them to freely recall their professional experiences in FGM/C fieldwork. The researcher occasionally posed follow-up questions for clarification and to dispel misunderstandings or specify certain concepts. This ensured a thorough understanding of the context referred to and maintained data accuracy. The interviews were recorded, with field notes taken during and at the end of each session.

### 2.3. Data Analysis

All the interviews were transcribed verbatim using the NVivo 14 transcription software [28]. Subsequently, each interview document underwent a thorough review alongside the corresponding audio recordings to ensure accuracy in the transcription and linguistic fidelity. The construction of the conceptual framework for the thematic analysis was based on the theoretical approach of Braun and Clarke. The thematic analysis technique involves identifying, understanding, and reporting patterns (themes) within the dataset (p. 79) [29]. This method is useful when attempting to comprehend experiences, thoughts, or behaviors across a dataset [30]. The data analysis followed the Braun and Clarke six-phase reflective thematic analysis protocol [30,31].

### 2.4. Coding

Our study employed a deductive approach to analyze the themes in the dataset; the codes and themes were informed by a pre-existing codebook (Table A1 in Appendix A) of the variables of interest linked with FGM/C cessation in a post-migration context. The author (N.S.) and the co-author (N.C.) read through the data and assigned citations to the pre-existing factors (themes) and sub-categories of factors (codes) associated with a decrease in support for FGM/C until data saturation was achieved. This study adopted a semantic approach within the thematic analysis, wherein the themes were identified within the explicit or surface meanings of the data. Ultimately, the thematic analysis in this study followed an essentialist or realist method; we reported the experiences, meanings, and reality of the participants as we sought to understand the knowledge of the professionals in the field of FGM/C [29]. The initial findings were discussed between the two researchers and a consensus was reached regarding the codes, the theme names, and selected quotes. No participants provided feedback on the findings.

### 2.5. Ethical Considerations

The participants were provided with a written consent form outlining the study’s objectives prior to the interviews. Permission was sought for the audio recording and, in some cases, for both the audio and video recording of the interviews. All the participants signed a written consent form, which was subsequently stored in a secure cupboard within the research facility of the department. To protect the participants’ anonymity, they were assigned a number after the interview. Confidentiality was maintained by deleting the interview records after data transcription. The project received approval from the Cantonal Research Ethics Commission. No compensation was provided to the study participants. 

## 3. Results

From the data analysis, four themes based on the ecological model of behavior change emerged, along with seven subcategories that encompass interconnected factors within and across dimensions (Table 1). The quotations selected to reflect these themes and factors were translated from French into English for greater clarity, as provided below. 

### 3.1. Social Level of Ecological Model of Behavior Change

#### Impact and Implementation of Swiss Anti-FGM/C Law

In our analysis, the initial dimension of the ecological model focusing on the social factors influencing changes in behavior toward FGM/C was associated with the effects and implementation of Swiss anti-FGM/C legislation. The participants mentioned that some individuals were aware of the anti-FGM/C laws in their home countries before migration. For example, according to one participant, young women from Eritrea had found that the practice was diminishing in their home country, and there was a general understanding that it was no longer practiced. The implementation of Article 124 of the Swiss Criminal Code was a key subject during the interviews. Two participants even mentioned that women were well aware of the existence of the Swiss anti-FGM/C law and that sometimes there was too much emphasis on the law, making it redundant for women with FGM/C. This view was reinforced by another participant:


*“That’s the difference with countries of origin where, in most cases, the law exists but is not enforced. What deters (FGM/C) in Switzerland is not only the existence of the law but, above all, its enforcement, and it is enforced extremely harshly.”*
(Participant 6)

The participant mentioned a criminal case in the canton of Neuchâtel in 2018, where a Somali woman was convicted of practicing FGM/C on her children; these acts occurred before their arrival in Switzerland. Some participants expressed concern that this event highlighted the challenges involved in conveying the protective intent of the Swiss law and its potential counterproductive effects. 


*“But I think there is pressure from the way the law has been applied because there were (court) appeals, and it didn’t succeed.”*
(Participant 2)

### 3.2. Community Level of Ecological Model of Behavior Change

#### 3.2.1. Religious Influence in Context of Migration

In our analysis, we found that the second dimension of the ecological model, which was centered on the community level, was correlated with four factors: (1) the influence of religion, (2) the social complexities associated with the practice, (3) the addressing of language barriers and the promotion of awareness of FGM/C, and (4) the cultural adaptation process. The primary sub-category showed that the perception of female genital mutilation is primarily linked to social factors rather than religious beliefs. One participant pointed to the Zurich trial in 2008 as an example of a shift in perception regarding FGM/C. Initially, a mother associated FGM/C with Islamic requirements. However, after interacting with Muslim women from various communities in Switzerland, she discovered that they were not circumcised, prompting her to question the connection between religion and FGM/C. 

The participants unanimously conveyed the view that the primary justification for the practice was rooted in the need to preserve cultural connections, landmarks, and traditions.


*“No patient talks to me about religion; she talks about tradition, not religion”*
(Participant 4)

However, another participant explained that some individuals, particularly those who originated from remote rural villages and those who had arrived in Switzerland with limited knowledge of religion, may perceive FGM/C through a religious lens rather than a cultural one.

#### 3.2.2. Multifaceted Examination of Social Aspects

The second sub-category of this dimension was associated with various aspects of social norms and values. Parents often engage in the practice in their home countries due to the prevailing cultural norms and beliefs linked to it. According to the participants, parents may perceive the practice positively, viewing it as beneficial for their children by fostering integration into the community and securing their future. In such situations, the procedure is crucial in achieving social acceptance and establishing a personal identity.


*“It was a source of pride to be circumcised. It was also the family’s pride, actually, to say, “Well, yes, I am a circumcised girl, I am a closed girl, of course, who is completely infibulated. Yes, exactly that.”*
(Participant 5)

In the migrant context, the participants noted that individuals do not seem to face pressure to conform to these norms and values or to undergo circumcision. One participant mentioned that there may be some pressure from the older generation, including grandmothers and mothers-in-law, but these sources of influence reside outside Switzerland. This is considered less influential than when the person is physically present.


*“Personally, I don’t feel that they (parents) are under pressure in Switzerland. In Switzerland, they are very clear; they are protected.”*
(Participant 4)

The participants mentioned that many immigrants face challenges in the host country, including difficulties with language and in navigating the medical system; these difficulties impede their adaptation to their new surroundings. Social precarity and uncertainty about the future can contribute to an increase in identity and cultural retreat. Nevertheless, one participant explained that young women are now encouraged to attain social and economic independence by themselves, rather than through marriage. This shift reflects the evolving mindset of today’s post-migration mothers, who focus on education, employment, and a future beyond traditional cultural expectations.


*“Nowadays, it’s about studying, working, having a job. At least, that’s what mothers think these days.”*
(Participant 5)

#### 3.2.3. Navigating Language Barriers and Awareness Raising

The third sub-category identified at the community level is related to the navigation of the challenges involved in language acquisition and to raising awareness about FGM/C. A participant working closely with migrant women, especially in language class settings, noted that navigating the challenges posed by language learning and integration can be particularly discouraging, as language acquisition remains a persistent difficulty for some women.


*“At the same time, what we also know is that breaking away from expectations is more challenging for a woman who has not been well-educated, thus a less-educated person. Moreover, it is a cultural objective in many situations to keep women in ignorance, as it allows better control over them.”*
(Participant 1)

Interestingly, another participant highlighted the fact that less-educated women also generally expressed a negative attitude toward the FGM/C practice. She reinforced her perspective by stating that women who displayed a good level of language proficiency were characterized by a certain sophistication in their responses regarding the practice, indicating a nuanced understanding of the conveyed message. These views stand in contrast to the perspective of a participant who expressed skepticism about education as a means of shifting views about the practice and questioned the widely held belief that education alone can eradicate FGM/C. However, many participants stated that educational tools and awareness campaigns were useful ways to inform communities. 


*“I accompanied a group, engaging in activities like working with modelling clay (representing male and female genitalia). For communities, particularly (with the help of) the intercultural mediators, this experience was foundational.”*
(Participant 1)

Despite the limited knowledge of the vulva’s anatomy, which is also a common problem in the general Swiss population, some participants were aware of the specific type of FGM/C that they had undergone. 


*“They knew themself. I don’t have the impression that they knew (the types of excision). I thought I remembered that they discovered a lot of things.”*
(Participant 2)

Another participant stated,


*“Some women, yes, they know what type of excision they have had, where they had it. Well, I had the pharaonic excision where everything is closed.”*
(Participant 5)

The participants also noted the absence of accessible information on a large scale for communities regarding prevention.


*“When it comes to the potentially affected population in Geneva in general, information is lacking.”*
(Participant 6)

#### 3.2.4. Cultural Adaptation Process

The fourth and final sub-category was connected to the process of cultural adaptation within the migration context. The participants stated that migration strengthened certain aspects, providing a sense of anchoring despite the experience of becoming a stranger in multiple places. FGM/C is perceived as potentially creating a connection with the country of origin, particularly for those who have recently migrated to Switzerland. 


*“It was clear that one had to take into account the length of stay because for them, a person who is not part of an asylum program, whose entire future is uncertain in fact, and who is trying to maintain a cultural link with their country of origin, may still find meaning in circumcision.”*
(Participant 6)

Furthermore, the participants suggested that the absence of excision cases in the communities could be attributable to a shift in mentality, as noted by cultural mediators working closely with the communities. This change does not necessarily occur post-migration; it can even manifest beforehand, in the country of origin. The migration process—especially migration to countries such as Switzerland, where excision is not practiced—plays a role in expediting this shift, leading to a loss of the significance attached to the practice.


*“I met women, whether they were directly affected, intercultural mediators, or second-generation descendants, who told me that they didn’t know anyone their age who had undergone the practice.”*
(Participant 6)

Another participant stated, 


*“Because the girls who are born here are in their twenties. They are no longer circumcised, even if they have daughters, they do not practice excision.”*
(Participant 5)

This can occur even in the case of asylum seekers, and one participant emphasized the impact of cultural adaptation, which countered any consideration of FGM/C. This influence acts as a deterrent and ultimately leads to the abandonment of the practice. This is because excision is no longer perceived as a guarantee of a young girl’s future. It no longer conforms to social norms; it lacks value, and it is no longer mandatory; in fact, it even poses difficulties in this new setting. 


*“Therefore, excision itself, I think, no longer has value here in Switzerland.”*
(Participant 4)

### 3.3. Interpersonal Level of Ecological Model

#### Changing Views on Marriage Prerequisite

We identified one sub-category within the ecological model in the interpersonal dimension; it was related to shifts in perspectives regarding marriage requirements. The participants noted that the concepts of purity, cleanliness, and hygiene were often cited to explain the reasons for the practice. According to one participant’s observations, when the women are interviewed there is an emphasis on preserving premarital virginity as a reason to continue the practice. Another participant noted that girls who arrive in Switzerland as unaccompanied minors or young adults may harbor a deep-seated belief that undergoing premarital defibulation (the reconstruction of the vaginal vestibule) will negatively impact their prospects of marriage. 


*“They can change the perception, but not change themselves. In fact, she will talk about it, she will discuss it, but she will think to herself, ‘if you take steps, well, it will affect you; that’s it; it will affect your marriage’.”*
(Participant 5)

In contrast, another participant pointed out that the significance of this practice diminishes with migration and that girls who have not undergone the procedure can marry in Switzerland. One participant mentioned that young Somalis or Eritreans may have connections with European women before marriage, either from their own community or elsewhere, and discover that the practice is not necessary.


*“I can’t believe that it’s still a requirement for young girls not to be circumcised, and that it would be the young men who would demand it for marriage. So I can’t conceive of it.”*
(Participant 6)

Another participant mentioned that, in her practice, she often encountered women who opposed FGM/C, considering it to have no value, especially in the context of marriage. 


*“The mothers I meet in consultations often do not share this viewpoint (as a requirement for marriage)”.*
(Participant 3)

### 3.4. Personal Level of Ecological Model of Behavior Change

#### Women’s Experience of and Perspectives on FGM/C

In our analysis, we observed that, within the personal dimension of the ecological model of behavior change, women’s individual experiences of and perspectives on FGM/C emerged. Personal viewpoints regarding FGM/C can play a significant role in changing attitudes toward the practice. Recalling her early career in the 1990s, one participant remarked that, while providing academic support, she observed academic difficulties among some Somali children. Her inquiries about women’s childbirth experiences revealed difficult deliveries involving fetal distress, especially for the firstborn, who may then encounter academic challenges.


*“He (the child) was born with more learning difficulties. A mother hears about this, and it resonates deeply, and she doesn’t want that. So, it was always a quite strong realization”.*
(Participant 1)

During pregnancy and postpartum consultations, couples expressed their opposition to performing FGM/C on their daughters. Despite resistance from some individuals, particularly those from rural areas and newcomers, mothers who had given birth in Switzerland viewed the practice as forbidden. This understanding was notably stronger among mothers who insisted on not sending their daughters back to their home countries, as they were aware of the potential risks for their children. 


*“I think that over all these years, I had one or two cases where there was suspicion that the family had left to perform excision. But it’s not confirmed. Two patients out of the thousands I see each year. It’s very few.”*
(Participant 3)

One participant stated that, upon arrival, women are informed by health professionals about the legal consequences and sanctions regarding the practice in Switzerland and the negative health consequences. 

The participants specified that, in Geneva, there is specialized care that enables a more sensitive approach to the addressing of the subject and offers educational tools such as brochures, informational materials, and awareness without preconceptions. 

*“I had mothers where I provoked them a bit. So you will leave (to your country). It (FGM/C) will be done? Patient: Oh no, it’s forbidden here”*.(Participant 3)

Referring to medical consultations, the participants recalled instances in which some women expressed physical pain, such as difficulties in urination and menstrual issues. During such medical consultations, these women may experience distress when discussing the practice, expressing emotions such as anger or frustration about what was done to them, especially when the conversation revolves around their children. Some women strongly oppose the practice, asserting that they will never subject their children to it and emphasizing the need for this harmful practice to cease. 


*“But it’s true that deep down, it’s something that hurts, that lingers, even if you’re 70 or 80 years old; it’s something that remains (…) My grandmother used to talk about it. She said, ‘But it was a suffering for us’.”*
(Participant 5)

One participant mentioned a conversation with a woman who expressed discomfort because the sound of her urination was audible after defibulation. However, the participant noted that most women in her medical practice felt better after undergoing defibulation surgery.


*“I still have the impression that there is a change in the perception of female circumcision, in mentalities, and in altering the woman’s body; it is changing, especially since people are aware that it causes more problems than anything else.”*
(Participant 4)

## 4. Discussion

Our findings highlighted four influential dimensions linked to the ecological model of behavior change, spanning the social, community, interpersonal, and personal levels [10]. Each dimension was further divided into seven sub-categories that encompassed the dynamic interrelationships between human behavior and environmental influences. These seven factors included (1) the impact and implementation of anti-FGM/C laws, (2) religion, (3) a multifaceted examination of social aspects, (4) the navigation of language barriers and the raising of awareness, (5) the cultural adaptation process, (6) changing views on the marriage prerequisite, and (7) women’s experiences of and perspectives on FGM/C. Berg et al.’s study highlighted that, in the post-migrant context, there is a shift in attitudes toward FGM/C, reflecting an evolving and dynamic cultural tradition that is described as a “tradition in transition” [32]. This transformation occurs due to selective migration and cultural change, where individuals with distinct characteristics tend to migrate and to gradually adapt to new norms and social values in their new country [12,14]. One of the primary observations made by the participants in our study was the discontinuation of FGM/C within the Geneva diaspora communities they engaged with.

Our study findings align with the previous research, which indicates a decline in support for FGM/C within post-migration communities in Western countries [13,16,17,33]. The first aspect at the social level of the ecological model suggests that there is a need to adjust the overall discourse surrounding FGM/C to align with the attitudinal changes occurring in communities. The participants highlighted the impact of Article 124 of the Swiss Criminal Code, enacted in 2012, which criminalizes FGM/C within Switzerland, particularly the universality clause allowing individuals to be prosecuted under Swiss law for FGM/C offenses committed in their home country [34]. The application of the law without properly clarifying the circumstances of the offense could result in communities facing disproportionate penalties due to the criminalization of FGM/C. It is important to move away from solely relying on criminal prosecution to address FGM/C or automatically assuming the existence of a risk for girls based solely on their ethnicity, or because they have a mother who has experienced FGM/C [34,35]. 

At the community level of the ecological model, four factors interact within this dimension, including religion, the influence of social aspects, barriers associated with language acquisition, and the impact of the cultural adaptation process. In our study, participants explained that individuals engage in the practice in their home countries due to prevailing social norms and values, which prioritize factors such as aesthetic considerations, reduced sexual arousal, preservation of virginity, and personal identity [36]. In the study by Kaplan et al., it was noted that individuals undergoing the process of abandoning cultural practices must decide on the elements of their culture to retain or discard as part of an adaptive strategy to fit into the new post-migration environment [37]. 

Participants highlighted that the motivations for undergoing FGM/C rarely revolved around religious considerations, but around customary or cultural ones [17]. While some individuals may see FGM/C as a way to preserve cultural connections and traditions, this perception does not necessarily mean they will continue the practice after migration [34]. This aspect was observed by participants when young women believed that if premarital defibulation was performed on them, it would negatively impact their chances for marriage and virtue. Another aspect emphasized in this level by participants is that some women face distinct challenges in the host country. This is particularly evident in language acquisition, which can persist as an issue, making it difficult for individuals to express their concerns. Moreover, women who possess proficient language skills are more likely to voice their concerns or seek care compared to those encountering communication barriers. In the study by Barrio-Ruiz et al., the absence of language proficiency hindered women in navigating the healthcare system [38]. Additionally, incomplete knowledge about available care fostered negative sentiments that could compromise access to adequate healthcare provision. Similarly, in the study by Evans et al., informal interpreters (such as family members) may only translate half of what was communicated during medical care, leading to lower trust and weakening effective healthcare provision [39]. Participants consistently emphasize the necessity for accessible, culturally sensitive information and awareness-raising regarding all aspects of FGM/C-related information and advocating for early dissemination in communities. 

The final aspect to consider at the community level is that the process of cultural adaptation is influenced by interconnected environmental factors [32]. In an environment where the practice is unfamiliar, with less social pressure to conform to FGM/C, individuals may be prompted to abandon it. For instance, participants mentioned that newly arrived individuals in the host country may prioritize pressing concerns such as legal status, housing, employment, and societal integration over considerations regarding FGM/C. Individuals may integrate through various avenues in the new society, such as family or education [14]. Individuals may reevaluate their views on FGM/C to align with the new norms and values prevailing in the host society. The combined effects of these factors may accelerate the shift and result in a decreased significance associated with the practice in post-migrant communities [13,16,17,33].

The important element highlighted at the interpersonal level of the ecological model was parents’ commitment to safeguarding their children’s well-being and their recognition of the potential physical and psychological health risks associated with FGM/C. Families express their commitment to protecting their daughters from the practice of FGM/C, regardless of whether they have faced any complications associated with it. Participants noted that mothers who have given birth in Switzerland unequivocally regard the practice as forbidden. They have decided not to send their daughters back to their home countries in order to protect them from harm [40]. This is underscored by participants’ observations that FGM/C loses its significance because it is no longer a prerequisite for marriage in the diaspora. Mothers encourage young women to pursue social and economic independence. This transition reflects the changing mindset within certain diaspora communities, envisioning a future beyond traditional cultural expectations for girls and women [40]. 

At the personal level of the ecological model, participants noted a decline in the social significance of the practice, with some even questioning its relevance entirely. Our findings, consistent with other studies, reveal that women challenged prevailing social norms and values associated with FGM/C by confronting the taboo surrounding it. Participants mentioned that discussions about FGM/C remained challenging for some women, especially with certain partners or family members from their home countries. However, women who were informed about FGM/C and were aware of its negative consequences began to critically examine the practice. By engaging in discussions and forming new social networks with other women who shared similar backgrounds, they started to reconsider internalized gender roles, compare their experiences, better resist external pressures through solidarity networks, and feel encouraged to take action against FGM/C. This process can promote individual and collective attitudinal change [20,41]. Overall, all participants in our study agreed that there is a gradual change in mindset in some communities, with clear signs that the practice is not continuing in the Swiss context.

### Strengths and Limitations

To the best of our knowledge, this study represents the first investigation that explores the multidimensional factors influencing attitudinal shifts within a post-migrant context. However, several limitations should be acknowledged. Our sample size was small, and the purposeful sampling method lacked representativeness; thus, the generalizability of our findings was restricted. Despite the small sample size, the participants brought extensive expertise from various FGM/C-related domains. Although these participants provided valuable insights into the FGM/C landscape in Geneva, their expertise cannot be generalized to other professionals in Switzerland. Furthermore, the Geneva context, with its specialized outpatient clinics for FGM/C care and highly trained professionals, may have biased the results toward favoring FGM/C cessation due to the patients being well informed about FGM/C upon arrival and to the social desirability of conforming to the prevailing anti-FGM/C attitudes in this context. 

## 5. Conclusions

Our study offers valuable insights into the perspectives of professionals in the field of FGM/C by examining various factors that influence views, beliefs, and attitudes toward post-migration communities. Our findings underscore the importance of providing clear preventive messaging on the legal, health-related, and social consequences of the practice; it can help individuals adjust their perceptions within a supportive environment. Embracing policies that reduce health and social barriers, promote socioeconomic independence, and employ empowering strategies in communities is essential. FGM/C policies should be framed within existing national strategies and developed collaboratively with various stakeholders to convey preventive messages in a culturally sensitive manner. Efforts should be made to create and adapt discourses aimed at destigmatizing communities, ensuring clear and accessible FGM/C risk assessment and reporting guidelines, professional training, and access to trained FGM/C community representatives. 

These efforts should prioritize both immediate actions and long-term investments to promote behavioral change, encouraging policymakers to allocate sustained funding for social and public health initiatives. Future research should explore additional potential factors influencing attitude changes within the diaspora, such as the role of men in preventing FGM/C or the preference for practicing less severe forms of FGM/C to preserve cultural identity. It should also focus on finding ways to achieve cultural preservation without resorting to FGM/C.

## Figures and Tables

**Table 1 ijerph-21-00716-t001:** Four dimensions associated with ecological model of behavior change.

Theme (Dimension)	Sub-Category (Factor)	Quotation Example
**1. Social level**	Impact and implementation of the Swiss anti-FGM/C law	“It’s not really that which will dissuade people. Because in most countries of origin, there is also a law, but that doesn’t prevent people from still committing female genital mutilation. In fact, where it changes, however, is when one knows that the law will be enforced.” (Participant 6)
**2. Community level**	Religious influence in the context of migration	“No, there is no patient who talks to me about religion. They talk about tradition, but they don’t mention religion to me.” (Participant 4)
	Multifaceted examination of social aspects	“There is no pressure in Switzerland. However, there can indeed be pressure from the grandmother, from the older generation in fact, and the mother-in-law as well, but these are individuals who do not live in Switzerland.” (Participant 6) “Nowadays, it’s about studying, working, having a job. At least, that’s what mothers think these days.” (Participant 5)
	Navigation of language barriers and raising of awareness	“The language can remain something difficult.” (Participant 1) “They knew themself. I don’t have the impression that they knew (the different types of excision). I thought I remembered that they discovered a lot of things.” (Participant 2)
	Cultural adaptation process	“The process of cultural adaptation is such that, the longer one lives in a country where female genital mutilation is no longer a social norm, is no longer obligatory. It is no longer a guarantee of a good future for the young girl, as it holds no value.” (Participant 6)
**3. Interpersonal level**	Changing views on the marriage prerequisite	“The significance attached to this practice is diminishing. In fact, a young girl can marry in Switzerland even if she is not circumcised.” (Participant 6)
**4. Personal level**	Women’s experiences of and perspectives on FGM/C	“I had women who said that they do not want their daughters to undergo the same thing, as they themselves had complications such as bleeding and infections.” (Participant 3) “I think that over all these years, I had one or two cases where there was suspicion that the family had left to perform excision. But it’s not confirmed. Two patients out of the thousands I see each year. It’s very few.” (Participant)

## Data Availability

The original contributions presented in the study are included in the article, further inquiries can be directed to the corresponding author/s.

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
