# Peer review of "Understanding Professionals’ Knowledge Regarding Factors Influencing Changes in Attitudes toward Female Genital Mutilation/Cutting in Post-Migration Communities in Geneva, Switzerland"

_ijerph, 2024, doi:10.3390/ijerph21060716_

Round 1
Reviewer 1 Report
Comments and Suggestions for Authors
this survey is a challenging task. The amount of data from the interviews is small, but it shows a representative selection
Comments on the Quality of English Languagethe language is difficult to understand. especially since the facts are difficult to describe
Author Response
We sincerely appreciate the time you dedicated to reviewing our article. Thank you for your valuable feedback on our manuscript.
Feedback 1
This survey is a challenging task. The amount of data from the interviews is small, but it shows a representative selection.
Comments on the Quality of English Language: The language is difficult to understand. especially since the facts are difficult to describe.
Response 1
In response to your feedback, we have revised the structure of our manuscript and reworked various sections of the article to enhance clarity. Specifically, we have reworded the introduction from line 37 to 101, streamlined the subsections, and interconnected the different theoretical frameworks. Additionally, we have rewritten the methodology section from line 105 to 175, reducing the overall length by 1 page. Finally, we have improved the readability of the discussion from line 408 to 507. In this section, we have segmented the different findings according to the four dimensions of the ecological model and emphasized key aspects, such as "The first aspect at the social level of the ecological model suggests that there is a need to adjust the overall discourse surrounding FGM/C to align with the attitudinal changes occurring in communities," as seen in lines 427-429. Overall, these revised versions have reduced the length of the manuscript. To complete the revision process, the paper was sent to an English editing service, conducted by a native English speaker. This step ensures readability, clarity, and understanding of the paper. We have incorporated all the suggestions and corrections provided in the editing feedback into our paper.
Reviewer 2 Report
Comments and Suggestions for Authors
Dear authors,
Congratulations on a well written and very important study. Although this was overall a well written article, I feel that the discussion section does not include a discussion and literature control of all the findings. Some of the factors are not included in the discussion. I would recommend that the authors do a critical reading of the discussion just to make sure that all the factors are referred to and discussed. There are also recommendations integrated into the discussion section and I was wondering, depending on the guidelines of the article, if this must not be done under a separate heading as it gets lost in the discussion. This is just a recommendation.
Comments on the Quality of English LanguageThe quality of English is good, there are minor editing required.
Author Response
We sincerely appreciate the time you dedicated to reviewing our article. Thank you for your valuable feedback on our manuscript.
Feedback 1
I feel that the discussion section does not include a discussion and literature control of all the findings. Some of the factors are not included in the discussion. I would recommend that the authors do a critical reading of the discussion just to make sure that all the factors are referred to and discussed.
Response 1
In response to your feedback, we've enhanced the discussion section's readability by organizing the various findings according to the four dimensions of the ecological model and emphasizing key aspects derived from the results section. To achieve this, we divided each section of the ecological model – social, community, interpersonal, and personal levels – into separate paragraphs, detailing associated factors within each section. For instance, we discussed how the first aspect at the social level underscores the necessity to align the discourse on FGM/C with changing attitudes in communities, citing the impact of Article 124 of the Swiss Criminal Code as highlighted by participants (as illustrated in lines 427-437). By organizing the discussion in this manner, we ensure that we don't overlook referencing any factors found in the results. Additionally, it allows us to demonstrate the interconnectedness between the ecological model and the environmental factors associated with changes in attitudes towards FGM/C in the post-migration context.
Feedback 2
There are also recommendations integrated into the discussion section and I was wondering, depending on the guidelines of the article. This must not be done under a separate heading as it gets lost in the discussion.
Response 2
In response to your recommendation, we have relocated elements in the conclusion section that refer to recommendations in light of our findings, spanning from line 524 to 542.
Reviewer 3 Report
Comments and Suggestions for Authors
Dear Authors,
I wish to express my gratitude for the opportunity provided by the journal to review your manuscript titled "Understanding Professionals’ Knowledge Regarding Factors Influencing Changes in Attitudes towards Female Genital Mutilation/Cutting (FGM/C) in Post-Migration Communities in Geneva, Switzerland." The issue of female genital mutilation represents a global public health challenge and a pertinent medicolegal topic due to its significant political and health implications. I appreciate your commitment to addressing this issue.
However, I must express some concerns that limit my ability to review your manuscript with the necessary attention. The text is excessively verbose and challenging to read, which may compromise the achievement of the study's objectives.
Specifically, I urge you to focus on the primary objective of the study and to summarize the content more clearly and concisely. Some necessary revisions include:
- Title: The acronym "FGM/C" must be spelled out and not used in the title.
- Abstract: The section concerning the results needs to be rewritten for clarity.
- Introduction: This section is overly verbose and requires a more concise summary. The study's objective should be clearly defined at the end of the introduction rather than in the discussion.
- Methods: Section 2.2 should be removed from the methods. Subsections need to be reworked and synthesized to improve readability.
A minor of English language is required
Author Response
We sincerely appreciate the time you dedicated to reviewing our article. Thank you for your valuable feedback on our manuscript.
Feedback 1
Title: The acronym "FGM/C" must be spelled out and not used in the title.
Response 1
In response to your feedback, we have revised the title from line 2 to 5 to spell out the practice instead of using the acronym: "Understanding Professionals’ Knowledge Regarding Factors Influencing Changes in Attitudes towards Female Genital Mutilation/Cutting in Post-Migration Communities in Geneva, Switzerland."
Feedback 2
Abstract: The section concerning the results needs to be rewritten for clarity.
Response 2
Based on your feedback, we have clarified the results section in the abstract, covering lines 21 to 31. For instance, we have interconnected the four dimensions of the ecological model with the environmental factors that influence changes in attitudes and behavior towards FGM/C in the post-migration context. This has helped to understand and clarify which factors, both within and across dimensions, are associated with each level of the ecological model.
Feedback 3
Introduction: This section is overly verbose and requires a more concise summary. The study's objective should be clearly defined at the end of the introduction rather than in the discussion.
Response 3
We have reorganized the introduction section from line 37 to 101. First, we reorganized the subsections and interconnected the different elements of theoretical frameworks. Then, we clarified the objective of the study at line 99-101.
Feedback 4
Methods: Section 2.2 should be removed from the methods. Subsections need to be reworked and synthesized to improve readability.
Response 4
As recommended in your feedback, we have removed section 2.2 from the manuscript. Additionally, we have rewritten the methodology section from line 105 to 175. In this section, we simplified the steps followed during the methodological process, reducing the number of subcategories from 11 to 5. This has also resulted in a reduction in the overall length of this section by 1 page.